# Diet-Induced Hypothalamic Inflammation, Phoenixin, and Subsequent Precocious Puberty

**DOI:** 10.3390/nu13103460

**Published:** 2021-09-29

**Authors:** Georgios Valsamakis, Angeliki Arapaki, Dimitris Balafoutas, Evangelia Charmandari, Nikolaos F. Vlahos

**Affiliations:** 1Second University Department of Obstetrics and Gynecology, Aretaieion University Hospital, Athens Medical School, Ethnikon and Kapodistriakon University of Athens, 15233 Athens, Greece; aarapaki@gmail.com (A.A.); dbalafoutas@yahoo.com (D.B.); nfvlahos@gmail.com (N.F.V.); 2First University Department of Paediatrics, Aghia Sophia University Children’s Hospital, Athens Medical School, Ethnikon and Kapodistriakon University of Athens, 15233 Athens, Greece; evangelia.charmandari@googlemail.com

**Keywords:** high-fat/glycaemic-index diet, hypothalamic inflammation, phoenixin, precocious puberty

## Abstract

Recent studies have shown a rise in precocious puberty, especially in girls. At the same time, childhood obesity due to overnutrition and energy imbalance is rising too. Nutrition and fertility are currently facing major challenges in our societies, and are interconnected. Studies have shown that high-fat and/or high-glycaemic-index diet can cause hypothalamic inflammation and microglial activation. Molecular and animal studies reveal that microglial activation seems to produce and activate prostaglandins, neurotrophic factors activating GnRH (gonadotropin-releasing hormone expressing neurons), thus initiating precocious puberty. GnRH neurons’ mechanisms of excitability are not well understood. In this review, we study the phenomenon of the rise of precocious puberty, we examine the physiology of GnRH neurons, and we review the recent literature regarding the pathophysiological mechanisms that connect diet-induced hypothalamic inflammation and diet-induced phoenixin regulation with precocious puberty.

## 1. Introduction

Puberty is the process of physical changes through which a child’s body matures into an adult body capable of sexual reproduction, and is marked by maturation of the genital organs, development of sexual characteristics, acceleration of growth, changes in affect and, in females, the occurrence of menarche [1]. It is caused by the activation and increased secretion of gonadotropin-releasing hormone (GnRH) by the hypothalamus, which in turn activates the gonads to produce hormones [2]. Precocious puberty (PP) is defined as is the onset of secondary sexual attributes before the age of 8 years in girls and 9 years in boys. PP is categorized either as Central PP (CPP), when premature maturation of the hypothalamic-pituitary-gonadal axis is present or as Peripheral PP (PPP), when an excessive secretion of sex hormones is recorded, regardless of gonadotropin secretion [3]. The prevalence of PP is not well-documented. However, 0.2% of girls and less than 0.05% in boys seemed to have some form of PP in the Danish population [4], while another study in Spain estimates the yearly incidence of CPP to be between 0.02 and 1.07 per 100,000 individuals [5]. The most interesting finding is the decline in the average age of the onset of puberty. During the last 15 years a 12-month decline in the mean age at the onset of breast development in Danish girls [6] has been found, while similar findings are reported in both Greek and Turkish populations [7,8]. Apart from genetic factors [9], it seems that environmental factors such as obesity and diet have an effect on the aforementioned trend [10].

Childhood obesity contributes a major public health problem. Overweight/obesity rates have been dramatically increasing during the last 40 years in many European countries, exceeding 30% and 10% among children and adolescents [11]. The international literature reports that diet, among many other factors, has an important effect on obeisity during childhood and adolescence. Indeed, high glycaemic-index/high glycaemic-load diets and high-fat diets are correlated with being overweight/obesity in children and adolescents [12,13]. Furthermore, studies have shown that high glycaemic-index/high glycaemic-load diets and high-fat diets are able to produce hypothalamic inflammation within days from their initiation [14]. Hypothalamic inflammation has been found to appear earlier, before significant weight gain, with rapid activation of a complex network of cells [15]. Activation of hypothalamic microglia in obese mice is influenced by dietary composition and by fat- and gut-derived hormones, rather than by obesity or adiposity per se [16]. Obese individuals store excessive fat in both subcutaneous and visceral adipose tissue [17,18], as well as in other organs such as skeletal muscles, vessels, liver and pancreas (ectopic fat) [19]. It seems that obesity, through fat storage leads to low-grade chronic inflammation in both adipose and other tissues [20]. Ectopic fat deposits are also correlated with neuroinflammation, and specifically, hypothalamic inflammation, as reported. Fat, via inflammatory molecules is correlated with secretion of orexigenic neuropeptides in the hypothalamus [21,22].

After puberty, gonadotropin-releasing hormone (GnRH) is normally released from the hypothalamus in order to activate the pituitary gland for secretion of luteinizing hormone (LH) and follicle-stimulating hormone (FSH) [23] GnRH-expressing neurons are located in various regions of the hypothalamus, such as the medial septum, the organum vasculosum and the rostral preoptic area, forming a neural network close to other central regulators. This position lets the GnRH network be affected by various neuroendocrine and metabolic signals [24]. A diet rich in fat can directly activate orexigenic neuropeptide signalling in the hypothalamus, inducing an inflammatory status. In addition, inflammatory mediators have an effect on orexigenic neuropeptides in the hypothalamus and the GnRH network [25].

There seems to be a relationship between nutrition and age at puberty. According to a meta-analysis [26], high energy intake was significantly associated with early menarche. However, the research into the potential association between a high-glycaemic-index diet and/or a high-fat diet has not yet led to conclusive results [10]. The aim of this review is to explore how a high-glycaemic-index diet and/or a high-fat diet could have an association with precocious puberty in either girls or boys.

## 2. Physiology of Puberty

Puberty is a crucial and transitional period appearing in children. During puberty, secondary sexual characteristics appear, growth accelerates, and children evolve psychosocially [10,27,28]. Puberty depends **on** the activation of the hypothalamic-pituitary-gonadal (HPG) axis [29].

At birth, the HPG axis is shortly and suddenly activated as a result of the absence of placental steroids that suppressed the axis. Subsequently, this activation leads to increased production of steroidal hormones and possible breast development and pubic hair in girls, a situation that is called “mini-puberty of infancy” [30]. This activation of HPG axis takes place between the first week and approximately the first 6 months of life [31]. Testosterone and oestradiol are secreted by testis in boys and ovaries in girls, respectively, due to increased levels of LH and FSH [10,27].

After this period, the HPG axis becomes inactive until its reactivation in adolescence. Gonadotropin-releasing hormone (GnRH) is secreted following a pulsatile pattern from the hypothalamus, leading to the secretion of LH and FSH from the anterior pituitary gland. The aforementioned hormones, in conjunction with oestradiol and testosterone, promote oogenesis and spermatogenesis, respectively. According to the literature, the arcuate nucleus, an anatomical location (located in the mediobasal hypothalamus) and neuropeptides kisspeptin, neurokinin B and dynorphin A that are co-expressed by the KNDy neurons, are important for GnRH pulse secretion [32]. Kisspeptin, a metabolic hormone plays a very important role in this GnRH pulsatile secretion since it stimulates GnRH neurons via the cognate G_q/11_-coupled receptor (KISS1R). Kisspeptin nerval cells, located in both hypothalamic and extrahypothalamic areas, mediate the feedback of the axis by the sex steroids, leptin and prolactin [33]. The important role of kisspeptin in GnRH release has also been proven in experimental models in rodents. Kisspeptin, along with another metabolic hormone, leptin, are not only associated with precocious puberty but are affected by nutritional factors (both over- and under-nutrition) via an epigenetic pathway [24]. Nitric oxide, secreted by specific neuronal cells, also has an effect on neuroendocrine communication [34].

Puberty appears at 11.2 ± 1.1 and at 11.6 ± 1.1 years in girls and boys, respectively. The timing of puberty is of great importance, since earlier puberty is associated with greater risk for social isolation, early sexual behaviour, shorter height in adult life and psychiatric disorders in general [10].

## 3. Precocious Puberty

The onset of breast or pubic hair development (Tanner stage B2) before 8 years of age in girls, and the appearance of pubic hair development, testicular enlargement of more than 3 mL and genital development (Tanner stage G2) before 9 years of age in boys, define what is known as precocious puberty (PP) [35]. The mean age of appearance of puberty has decreased significantly over the last 100 years in Europe and worldwide. Mean age at menarche has been reduced from 17 years in the nineteenth century to approximately 12 years. According to the literature, the frequency of girls having PP ranges from 6.7% to 10.4% [36,37].

Precocious puberty is categorized as: a) Central precocious puberty (CPP) and b) Peripheral precocious puberty (PPP). CPP is caused by premature activation of the hypothalamic-pituitary-gonadal (HPG) axis, thus is GnRH dependent, while PPP is not [38].

### 3.1. Central Precocious Puberty (CPP)

The earlier-than-usual activation of the HPG axis leads subsequently to maturation and the appearance of puberty. CPP prevalence is estimated to be 10^−4^ and 10 times more frequent in girls than boys [38]. CPP aetiology includes multiple factors such as genetic factors (KISS1 gain-of-function mutations, KISS1R gain-of-function mutations, MKRN3 loss-of-function mutations, DLK1 loss-of-function mutations), syndromic factors (Prader–Willi syndrome, Pallister–Hall syndrome, Type 1 neurofibromatosis, Sturge–Weber syndrome), CNS tumours (Hypothalamic hamartoma, pineal cyst, glioma), CNS disorders (Brain injury, encephalopathy, hydrocephalus, neonatal infections, cerebral palsy) and environmental factors (early life stress, adoption, prepubertal exposures to sex steroids, nutritional disorders) [24].

### 3.2. Peripheral Precocious Puberty (PPP)

The earlier appearance of secondary sexual characteristics, not dependant on GnRH levels, constitutes PPP. This disorder derives from secretion of sex steroids from both internal and external sources and is rarer than CPP. Some documented causes of PPP are sex cord-stromal tumours (Leydig cell tumours, Sertoli cell tumours), germ-cell tumours (dysgerminoma, teratoma, and embryonal tumours), adrenal tumours, congenital adrenal hyperplasia, testitoxicosis, congenital adrenal hyperplasia, McCune–Albright syndrome, Van Wyk and Grumbach syndrome and exposure to exogenous sex steroids [39].

## 4. Potential Mechanisms of Diet-Induced Precocious Pubarche

Observatory data demonstrate that unhealthy diet patterns are significantly positively associated with precocious puberty in children [40]. There are two possible mechanisms:
(A)Activation of GnRH via Hypothalamic Microglial Activation (Figure 1)


High-fat diet causes hypothalamic inflammation and microglial activation. It has been shown that microglia of the hypothalamus are sensitive to fatty acids [41]. Interestingly, fatty acids, and not obesity per se, are required to induce hypothalamic microglial activation, since genetically obese mice do not present hypothalamic microglia activation in the absence of high-fat diet [16,42]. There exists evidence of intercellular communication, mediated by prostaglandins, from microglia to GnRH cells [43]. Prostaglandins seem to be an important mediator. Indeed, triple-labelled immunofluorescent histochemistry has proved the expression of the enzyme cyclooxygenase, which catalyses the synthesis of prostaglandins in ramified microglia in the vicinity of GnRH neurons. Additionally, further evidence suggests that microglia enhance the production of neurotrophic factors for several neurons [44].

A microglia–neuron interaction has been shown through the effect of brain-derived neurotrophic factor (BDNF), which controls neuronal excitability by causing disinhibition [45]. The synthesis and release of BDNF in microglia is mediated by the purinergic receptor P2X4R, an ATP-activated molecule which triggers morphological changes in microglia from a resting to an activated state [46]. Furthermore, treatment of BDNF on transgenic mice-derived cell-culture levels indicated a highly significant stimulatory effect of BDNF on GnRH primary neurite length [47]. Recently, experiments in sheep have demonstrated that central treatment with exogenous BDNF does stimulate GnRH mRNA in the preoptic area and in the pituitary [48].
(B)Activation of GnRH via diet-induced phoenixin action (Figure 2)

In addition, it is well established that central control of puberty is exercised through GnRH [49], however, the mechanism by which the GnRH neuronal network integrates developmental and metabolic factors is less understood. There are data for the upstream neuronal regulation of GnRH through the kisspeptin signalling pathway [50]. Indeed, neurons expressing this peptide in the hypothalamus seem to have a direct action on individual GnRH neurons, which is electrophysiologically characterized by a potent depolarization [51].

Another mediator which has been shown to activate GnRH and kisspeptin neurons is phoenixin. Phoenixin is a neuropeptide that seems to mediate anterior pituitary function in fertility and was recently described through information derived from the Human Genome Project [52]. There is evidence of phoenixin expression in various tissues, including the hypothalamus, and the peptide demonstrates high preservation among species, which underlines its physiological role. Phoenixin also exerts anti-inflammatory, and cell-protective effects, as shown in mice [53]. Additionally, binding sites in the ovaries have been reported, where there is evidence of expression of the phoenixin precursor molecule, SMIM20 [54]. There has been thorough investigation regarding the phoenixin signalling pathway and it has been shown that phoenixin activates immortalized GnRH and kisspeptin neurons through the GPR173 receptor [55,56].

Phoenixin induces gonadotropin secretion though GnRH stimulation mediated by kisspeptin, a hypothalamic neuropeptide which potently stimulates GnRH release [57,58]. Interestingly, the crucial role of kisspeptin for puberty is proven, because inactivating mutations of the kisspeptin receptor results in patients who fail to progress through puberty [59,60].

The direct stimulating effect of phoenixin and its receptor, GPR173, in human ovarian follicles has also been demonstrated [61] and phoenixin induces oestradiol production in a dose-dependent manner.

Additionally recent research has shown a possible regulation of phoenixin expression by fatty acids [62]. The expression of phoenixin at the hypothalamic level could allow it to be regulated by peripheral signals, such as hormones and fatty acids. The latter are known to cross the blood–brain barrier at the median eminence and regulate hypothalamic peptide expression [62]. Immortalized hypothalamic neurons treated with fatty acids (palmitate, DHA and oleate) led to increase in phoenixin mRNA levels measured with RT-qPCR. Therefore, dietary signals in the hypothalamus seem to exercise nutritional control of reproduction. In the same study, elevated levels of cAMP, NO and activation of PKC and neuroinflammation had no effect on phoenixin mRNA.

Another study showed that palmitate activates neuroinflammatory signalling [63], but its effects on phoenixin mRNA were independent of neuroinflammation, as treatment with lipopolysaccharide (LPS) did not change expression. According to these results, the effects of fatty acids on phoenixin seem to be in response to nutritional factors rather than inflammatory factors. Other data suggest that phoenixin-20 has a regulatory effect via modulation of neuroinflammation, ameliorating LPS-induced activation of microglial NLRP3 inflammasome [64,65], suggesting that phoenixin-20 possesses a protective effect against LPS-induced activation of the NLRP3 inflammasome in microglia.

## 5. Conclusions

Animal and molecular studies propose mechanisms on how diet-induced hypothalamic inflammation could be associated with precocious puberty. Indeed, they suggest that a high-fat and/or high-glycaemic-index diet induces a low-grade hypothalamic inflammation and subsequent premature GnRH activation. Potential mechanisms show that the above diet induces low-grade inflammation in the hypothalamus, and that the subsequent microglial activation mediates prostaglandins and neurotrophic factors to GnRH cells, thus activating precocious puberty. Furthermore, these studies show that fatty acids stimulate phoenixin secretion, a neuropeptide that activates GnRH neurons. Thus, diet induces phoenixin secretion, acting into GnRH neurons and leading to subsequent premature GnRH activation, whereas the neuron inflammation is modulated by phoenixin [59,60,61]. In addition, it is not clear whether the two mentioned mechanisms, e.g., 1. Diet-induced hypothalamic inflammation, microglial activation mediating prostaglandins and neurotrophic factors to GnRH cells and 2. Diet-induced phoenixin secretion, a neuropeptide that activates GnRH neurons, are all part of a sole process or are two separate mechanisms. Furthermore, the nature (anti-inflammatory?) of the above-described microglial activation reactions and their effect (protective?) on neuronal cells which leads to premature GnRH neuron activation is not clear yet. Therefore, the mechanisms and triggers of GnRH neuron activation are not well understood and represent an exciting area of future research.

The above studies serve as proposed mechanisms on how diet could activate the fertility axis. As nutrition is one of the major issues in our societies, and precocious puberty is increasing, especially in girls, it is important to introduce protocols in children that will study the effects of diet-induced hypothalamic inflammation on GnRH neuronal activation. Indeed, energy balance and macro/micronutrient constitution of food need further studying regarding their effect on GnRH neurons excitability. Additionally, there is a need for improvement in our methodology to approach and understand the pathophysiological and molecular mechanisms of hypothalamic inflammation and its effects on the central nervous system neuronal function.

## Figures and Tables

**Figure 1 nutrients-13-03460-f001:**
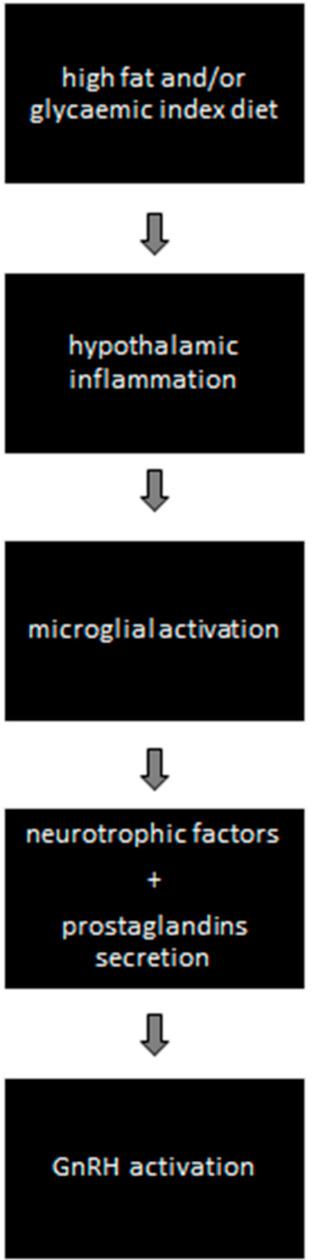
Diet-induced hypothalamic microglial activation and GnRH activation.

**Figure 2 nutrients-13-03460-f002:**
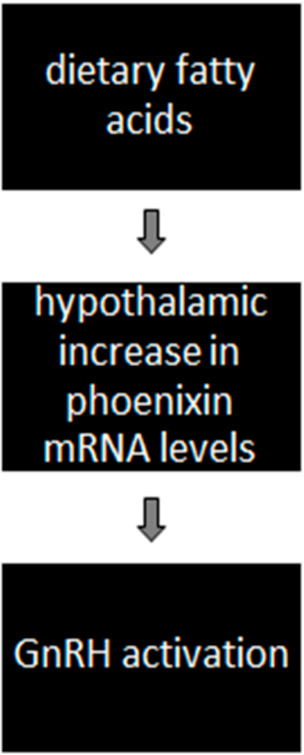
Diet-induced phoenixin mRNA upregulation and GnRH activation.

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
