# Peer review of "Diet-Induced Hypothalamic Inflammation, Phoenixin, and Subsequent Precocious Puberty"

_nutrients, 2021, doi:10.3390/nu13103460_

Round 1

Reviewer 1 Report

The manuscript is interesting and well written. The topic is extremely important in the light of the increase in the incidence of precocious puberty in recent years. The hypothesis considered by the authors of a direct link between a diet rich in fat and high glycemic index foods and hypothalamic inflammation with the consequent induction of precocious puberty is based on previous studies on the animal and must be confirmed in child studies. For this reason the statements in the conclusions must be rewritten avoiding drawing definitive conclusions but proposing a new mechanism of action that will have to be confirmed with other studies. 

Reviewer 2 Report

This is a very short review on how inflammatory mechanisms in the hypothalamus affect GnRH neurons and therefore lead to premature puberty. The authors give a short overview of the current literature, especially on the activation of GnRH neurons via hypothalamic microglial activation.

  • The authors cover the topic in broad strokes without going into too much detail. My biggest concern is that this review is extremely short, superficial and fails to critically penetrate the data out there.

  • The authors “review recent literature regarding pathophysiological mechanisms that connect diet induced hypothalamic inflammation with precocious puberty”. In chapter 4 the authors focus on two mechanisms:
  1. Activation of GnRH via hypothalamic microglial activation of porstaglandins and BDNF
  2. (Dietary) Activation of GnRH through phoenixin

My biggest issue with this two mechanisms is, that phoenixin as the authors write themselves “… is secreted as a result of local hypothalamic inflammation and subsequent microglial activation”.

Therefore, microglial activation seems to be the common mode of action in the activation of GnRH neurons (mentioned in A) and B)) and the two mechanisms as described in this review are redundant and could be incorporated into a single paragraph. Furthermore, please privde more information about the potential secretion of phoenixin from (activated) microglia.

  • Where is the chapter 3.2. “Peripheral precocious puberty (PPP)”? This chapter is completely missing. Please provide a detailed overview about peripheral precocious puberty!

  • The description of the findings is merely a list of citations with virtually no (re)interpretation, discussion, potential disagreement, and evaluation of gaps in the literature. A good review however provides a new conceptual framework, it penetrates and critically evaluates the data and discusses issues that we commonly do not have time to discuss in original publications. Here is where this review falls short!

  • At least one or two illustration(s) should be provided to give an overview about this new conceptual framework.

Here are some more points in random order:

  • Please use line numbers – this helps the reviewer to clearly point to issues in the text!
  • Please do not write “nerval cells” – use “neurons” (or “neuronal cells”)
  • Abstract – “glycaemic”, rest of the review: “glycemic”
  • In 2. “physiology of puberty” the authors mention arcuate nucleus (ARC) and kisspeptin/neurokinin B/dynorphin A (KNDy) neurons – Here, it is important to note that the arcuate nucleus is an anatomical location (located in the mediobasal hypothalamus) and kisspeptin, neurokinin B and dynorphin A are neuropeptides, that are co-expressed by the mentioned KNDy neurons. It is easy to confuse ARC as a neuropeptide.
  • In 5. “Conclusion”: Where do infections suddenly come from? They are not mentioned anywhere in the review!
  • In the abstract the second sentence is already about “GnRH neurons…”, without even describing the abbreviation. This sentence should be further down in the abstract, where the focus is on GnRH neurons.
  • In the first sentence of the Introduction – “Puberty … the process which humans transform from child to an adult”. Please rewrite – this sentence is not acceptable. Please use literature for a consensus definition of puberty.
  • In the introduction, the authors provide a paragraph on adipose tissue and ectopic lipid deposition. In the last sentence they mention that “hypothalamic inflammation has been found to appear earlier than overweight and obesity”. This is exactly the point – hypothalamic inflammation occurs for example within one day of high-fat diet feeding (see Jais et al., PMID: 28045396 for a review on the topic) and therefore much earlier than ectopic fat depots. The paragraph should be re-written to reflect this early and late effects during the development of obesity.
  • In Chapter 2 “Physiology of Puberty” – Citation for “mini-puberty of the infancy”?
  • Close to the end of the review – Consider writing “… study the effects of diet induced hypothalamic inflammation on GnRH neuronal activation”
  • The authors might rethink the last sentence of their review. Is this a really how you would like to end this review?
  • Proof-reading by a native english speaker would greatly improve this manuscript
